# The Lateralization of Resting Motor Threshold to Predict Medication-Mediated Improvement in Parkinson’s Disease

**DOI:** 10.3390/brainsci12070842

**Published:** 2022-06-28

**Authors:** Tomoo Mano, Kaoru Kinugawa, Shigekazu Fujimura, Kazuma Sugie

**Affiliations:** 1Department of Neurology, Nara Medical University, Nara 634-8521, Japan; kinugawa_kaoru@naramed-u.ac.jp (K.K.); ksugie@naramed-u.ac.jp (K.S.); 2Department of Rehabilitation Medicine, Nara Medical University, Nara 634-8521, Japan; shigers36style@naramed-u.ac.jp

**Keywords:** cortical excitability, motor threshold, Parkinson’s disease, resting motor thresholds, biomarker

## Abstract

Cortical stimulation patterns in patients with Parkinson’s disease (PD) are asymmetric and get altered over time. This study examined cortical neurophysiological markers for PD and identified neurophysiological markers for lateralization in PD. We used transcranial magnetic stimulation (TMS) to study corticospinal and intracortical excitability in 21 patients with idiopathic PD. We used the Movement Disorder Society Unified Parkinson’s Disease Rating Scale for examination during on and off periods and evaluated inhibitory and facilitatory process markers using TMS, including resting motor thresholds (RMT), active motor thresholds, and motor evoked potential amplitude. The RMT in the more affected cortex was significantly shorter than in the less affected cortex, and was strongly correlated with improved motor function following medication. Patients in the tremor group exhibited significantly lower RMT compared to those in the akinetic-rigid group. Cortical electrophysiological laterality observed in patients with PD may be a useful marker for guiding treatment and identifying underlying compensatory mechanisms.

## 1. Introduction

The average frequency of cortical oscillations in patients diagnosed with Parkinson’s disease (PD) decreases over the course of the disease [1]. The primary pathological feature of PD is the presence of intraneuronal proteinaceous inclusions, termed Lewy bodies, primarily composed of α-synuclein [2]. This trend in cortical excitability was consistently observed despite heterogeneity in disease stage at the time of examination, clinical presentation, and medication status [3]. The symptom lateralization is an important feature of PD, which has been investigated as a potential predictor of disease severity or progression. This symptom lateralization is diagnostically important and distinguishes PD from other motor disorders with overlapping symptom profiles. This clinical asymmetry corresponds to a neurochemical lateralization, and postmortem as well as in vivo studies have demonstrated the correlation of motor involvement with contralateral dopaminergic deficit [4].

Although drug adjustment is essential in PD treatment, there are still gaps in the evaluation of symptom improvement. We explored whether the fluctuation between on and off periods (hereafter ON and OFF, respectively) is related to cortical function, such as cortical inhibitory and facilitatory processes. 

Transcranial magnetic stimulation (TMS) is a valuable method to investigate the temporal dynamics of the motor system during perception of emotional faces, via stimulation of the primary motor cortex (M1) and the consequent induction of motor-evoked potentials (MEPs) in target muscles. The amplitudes of TMS-induced MEPs provide an instantaneous readout of excitability of the corticospinal system, allowing the probing of distinct motor representations with high temporal resolution, and importantly, to distinguish between excitatory (MEP increase) and inhibitory (MEP decrease) motor processes [5]. MEPs generally reflect the corticospinal input–output balance [6], and correlate with motor cortical excitability [5,7]. Various changes in TMS parameters, which exhibit differential effects across pathological conditions, have been reported in patients with PD [8]. The resting motor threshold (RMT) is the lowest stimulus intensity that can evoke MEPs. It has been hypothesized that RMT depends on the excitability of the associated neural elements, which are stimulated by TMS and subsequently propagate the elicited action potential [9]. 

We hypothesize that there are associations/relationships between cortical inhibition/facilitation and the disease severity, laterality of motor symptoms, and scope of response to drugs administered in PD treatment.

## 2. Materials and Methods

### 2.1. Participants

We selected patients who fulfilled the United Kingdom Parkinson’s Disease Society Brain Bank (UK-PDSBB) clinical diagnostic criteria for probable PD. All patients exhibited no significant medical, neurological, or psychiatric diseases or symptoms other than those of PD. We enrolled 29 patients with sporadic PD who visited Nara Medical University Hospital between May 2020 and February 2021. We included Japanese patients with confirmed PD who met the UK-PDSBB criteria. The age of the patients was between 20–90 years, and we included individuals who were capable of ambulatory hospital visits. The exclusion criteria were as follows: (1) history of neurosurgical operations, such as deep brain stimulation; (2) history of epilepsy; and (3) current use of any implantable heart stimulation or assist device, such as pacemakers. A total of 21 right-handed patients fulfilled the inclusion criteria, and eight patients were excluded (Table 1).

We evaluated the patients’ motor and non-motor symptoms using the Movement Disorder Society Unified Parkinson’s Disease Rating Scale (MDS-UPDRS). Patients with PD were observed for 1 week prior to examinations and were assessed in the ON and OFF states, which correspond to the states when dopamine replacement therapy is effective. Symptoms were controlled and when the treatment effect wore off and symptoms re-emerged. Motor symptom examinations were performed on each patient during both the ON and OFF states over two testing days by the same examiner. We identified the more affected side (MAS) as the laterality with higher tremor and rigidity scores than the contralateral side. Operationally, this was the side with the higher score on the nine-item MDS-UPDRS part 3; by default, the less affected side (LAS) was the laterality that scored lower on the same scale. MDS-UPDRS motor score was further divided according to PD lateralization in the MAS and LAS, obtained upon adding items 3.3–3.8 and 3.15–3.17, respectively, for each hemibody [10]. The axial symptom defined items 3.9–3.14. Next, the patients were classified as being akinetic/rigid-affected (PD-AR) or tremor-affected (PD-TD) based on the ratio of tremor scores and akinetic/rigid scores of the MDS-UPDRS [11].

We defined disease motor symptom onset as the time when muscle weakness or involuntary movement began rather than when non-motor symptoms appeared because non-motor symptoms may precede muscle weakness by many years in some patients with PD.

Ethics committee approval for the study was obtained from Nara Medical University. All study procedures were performed in accordance with the ethical standards of the committee, the 1975 Declaration of Helsinki, and Ethical Guidelines for Medical and Health Research Involving Human Subjects in Japan. All participants provided verbal and written informed consent after receiving information about the study.

### 2.2. Transcranial Magnetic Stimulation (TMS)

TMS was performed using a stimulator (MagPro X100, MagVenture, Farum, Denmark) that induced biphasic magnetic pulses via a figure-8 coil (MC-B70, MagVenture, Farum, Denmark). A single stimulus of progressively increasing intensity was applied to determine RMT for an MEP in the contralateral thenar muscles. RMT was defined as the lowest stimulus intensity that induced an MEP amplitude of at least 100 μV in at least 50% of a series of six stimuli while muscles were at rest. MEPs were recorded using Ag/AgCl surface electrodes in a belly-tendon montage. Electromyogram (EMG) signals were amplified with a 500–3000 Hz bandpass filter and digitally sampled at 10 kHz. Accurately determining RMT is important for patient safety; this reduces the number of TMS pulses delivered to a patient [12]. We recorded MEPs to five stimuli at approximately 120% RMT. Subsequently, we determined active motor threshold intensity (AMT) by asking the patient to push their thumb against their index finger (a sustained voluntary contraction) at approximately 20% of their maximal strength. AMT was defined as the lowest stimulus intensity capable of inducing an MEP recognizable above the background activity in at least 50% of the series. The contralateral side of MAS was defined as more impaired cortex, and the contralateral side of LAS was defined as less impaired cortex.

### 2.3. Data Analysis

Differences in categorical variables were assessed using the χ^2^ test. The Shapiro–Wilk test was used to assess the distribution of data. Correlations among the parameters were analyzed using Spearman’s correlation coefficient, and correlation coefficients (r) greater than 0.4 were defined as being significant. We used a *t*-test for comparison between the two groups, and *p*-values less than 0.05 were considered as significant difference. Calculations were performed using Statistical Package for the Social Sciences 22.0J (SPSS Japan, Tokyo, Japan).

## 3. Results

### 3.1. Clinical Backgrounds and TMS in Patients with PD

The clinical features of the patients are summarized in Table 1. Of the 21 patients, two required a cane for ambulation. No patient was bedridden or wheelchair-bound. The demographic and clinical characteristics of the subjects and their UPDRS III scores in the ON and OFF periods are presented in Table 1. The L-dopa equivalent dose (LEED) was not correlated with disease duration. Levels of RMT showed significant differences between the more affected cortex and less affected cortex in the OFF period (Figure 1), but not in the ON period. We investigated whether changes in TMS reflected the degree of disease progression. The level of RMT did not correlate with disease duration, Hoehn and Yahr scale (H-Y), LEED, and age.

Eight patients (four males and four females, aged 73.7 ± 7.3 years old, with disease duration of 9.3 ± 8.7 years) were designated PD-TD, and 12 patients (five male and four female, aged 72.0 ± 9.2 years old, with disease duration of 11.8 ± 7.7 years) were designated PD-AR. The levels of mean RMT in the RD-PD group were significantly higher than in the TD-PD group (*p* < 0.05), but MAS and LAS were not significantly different.

### 3.2. Correlation between Transcranial Magnetic Stimulation Parameter and Motor Symptoms

The associations between the transcranial magnetic stimulation parameter and the motor symptoms (MDS-UPDRS part 3) are summarized in Figure 2. The RMT in the more affected cortex was correlated with the MDS-UPDRS part 3 in the OFF period. However, the RMT in the less affected cortex did not exhibit this association. The RMTs of both the more affected cortex and less affected cortex were correlated with the MDS-UPDRS part 3 in the ON and OFF period. 

### 3.3. Correlation between RMT and Motor Subscore

We investigated the relationship between RMT and motor subscores in the axial, more affected side, and less affected side. In OFF period, the RMT in the more affected cortex was strongly correlated with motor symptoms of the axial and more affected sides, but not with the less affected side. In contrast, there was no correlation in the ON period (see Table 2).

### 3.4. Correlation between RMT and Motor Fluctuation between ON and OFF

Next, we explored the relationship between the change in RMT and motor fluctuation between the ON and OFF periods. The change in more affected and mean RMT between ON and OFF correlated with the fluctuation in UPDRS part 3, suggesting that the intensity of corticospinal neuron excitability between these periods is associated with motor fluctuation. The amount of change in motor fluctuation from the OFF period to ON varied inversely with RMT in the more affected cortex (Figure 3). This trend was not found in the less affected cortex.

## 4. Discussion

This study investigated whether RMT and AMT, which are associated with cortical inhibitory and facilitatory processes, reflect disease severity in patients with PD. Our results demonstrated that RMT strongly correlated with motor fluctuation between ON and OFF periods. This indicated that RMT in the more affected side may be a valuable parameter for determining medication-facilitated improvement. 

Dopamine depletion in PD causes not only an increase in concentration of gamma-aminobutyric acid (GABA) in the thalamus, but also a decrease in the promoting effect of the motor cortex. Some reports suggest that PD impairment is associated with elevated cortical beta power [13], while other studies have found that beta power is decreased in PD and is elevated by L-dopa administration [14]. Melgari et al. [15] found an increase in alpha and beta band EEG power with L-dopa that was correlated with improvement in rigidity and bradykinesia. This study showed that RMT levels correlated with motor function. RMT, which relates to resting membrane potential properties of corticospinal motor neurons, is associated with hyperexcitability of the motor cortex, which indicates that changes in GABA-B and glutamate function are important factors in cognitive impairment [16]. This activation–deactivation process has been hypothesized to be slow in PD, manifesting clinically as bradykinesia. This could correlate with motor dysfunction, particularly a compensatory physiological change, aiming at near-normal motor functioning [17]. A previous study reported that GABA levels in the motor cortex were inversely correlated with disease severity in patients with PD [18]. Another study showed that in early-stage PD, motor cortical beta power was higher, contralateral to the more affected side, and this asymmetry was reduced by a GABA-A receptor agonist [19].

RMT has been used as an index of corticospinal neuron excitability in a wide range of normal and disease periods [20]. The RMT is the most reliable parameter for assessing corticomotor excitability because it is less influenced by other factors [21]. A previous study [3] reported the difference in the TMS parameter between the more affected cortex and less affected cortex in hemiparkinsonian patients. The present study also showed RMT is the difference between the more affected cortex and less affected cortex, and associated with motor fluctuation in PD. The RMT in the more affected cortex can be considered as a potential biomarker for predicting fluctuations in motor dysfunction severity. Collectively, motor dysfunction is associated with corticomotor excitability in the more affected side of PD.

This study is the first to directly compare the laterality of motor-related corticomotor excitability responses with symptom asymmetry in patients with PD whose symptoms were one-sided. However, the study has several limitations. First, even in the OFF period, our patients showed symptoms, which likely influenced the amount of change observed. Therefore, the results should be interpreted with caution. This study also provides preliminary evidence regarding the neurophysiological basis of preferential behavioral outcomes in lateralization of PD [22]. However, the levels of TMS parameter were not inadequately considered, the floor and ceiling effects could not be adequately analyzed in this study. Further studies should evaluate potential treatment differences, and consider monitoring the course of disease progression in participants who exhibit uniform severity in the OFF period. Second, the patients in this study only showed right-handedness, which could have interfered with the results. It is necessary to include both types of patients, with left-handedness and right-handedness, in the study. Third, this study has a small sample size. Based on the obtained results, a large clinical trial should be performed to validate our findings in the appropriate populations. 

## Figures and Tables

**Figure 1 brainsci-12-00842-f001:**
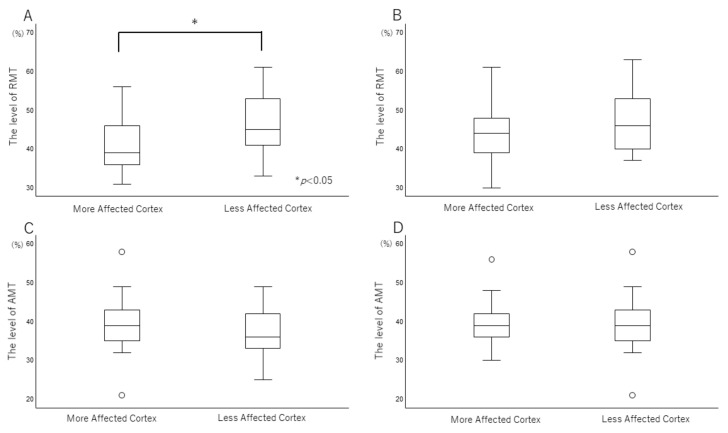
The comparison between more affected side and less affected side. (**A**,**C**) OFF period, (**B**,**D**) ON period. The levels of the RMT in the more affected side were significantly lower than those in the less affected side in OFF state (*p* < 0.01). Median values are represented by a thick black horizontal line within the box, and the box represents the upper and lower quartiles. The whiskers represent the maximum and minimum values, excluding outliers, which are shown as small circles, at least 1.5 times the interquartile range.

**Figure 2 brainsci-12-00842-f002:**
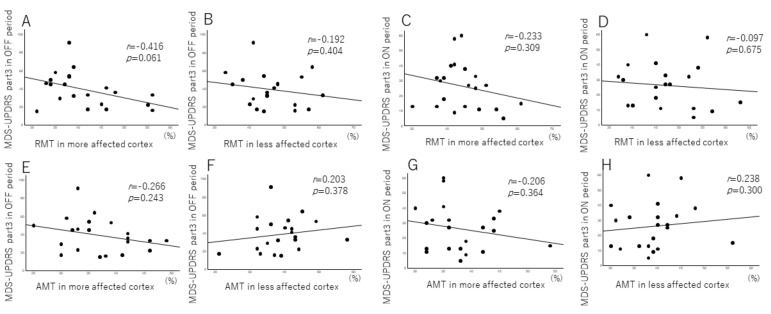
Correlation between the transcranial magnetic stimulation parameter and the motor symptoms as per the Movement Disorder Society Unified Parkinson’s Disease Rating Scale (MDS-UPDRS) part 3. (**A**) Resting motor thresholds (RMTs) in the more affected cortex were correlated with the MDS-UPDRS part 3 in the OFF period. (**B**) RMT in the less affected cortex was not correlated with the MDS-UPDRS part 3 in the OFF period. (**C**,**D**) RMT was not correlated with the MDS-UPDRS part 3 in the ON period. (**E**–**H**) Active motor thresholds (AMT) in more and less affected cortex were not correlated in the ON and OFF periods.

**Figure 3 brainsci-12-00842-f003:**
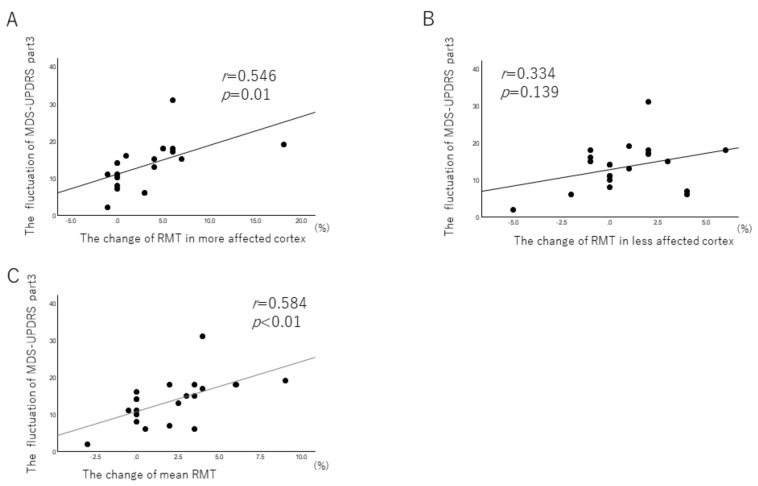
The relationship between the change in RMT and the motor fluctuation between ON and OFF in Parkinson’s disease. (**A**–**C**): (**A**), more affected cortex, (**B**), less affected cortex, (**C**) mean RMT.

**Table 1 brainsci-12-00842-t001:** Clinical features of patients with PD (*n* = 21).

		PD ON State	PD OFF State	*p*-Value
Number of subjects	21			
Sex	Female; 11, Male; 10			
Age at examination (years) *	72.7 ± 8.2			
Duration from motor symptom onset (years) *	10.7 ± 8.0 3.3 ± 0.86			
H-Y (years) *	693.1 ± 279.8			
LEED *		64.9 ± 30.6	81.5 ± 34.0	<0.05
MDS-UPDRS Total *	14.4 ± 7.6			
Part 1 *		16.6 ± 10.1	20.1 ± 10.8	<0.05
Part 2 *		26.2 ± 15.4	39.9 ± 19.1	<0.05
Part 3 *	7.1 ± 3.9			
Part 4 *	Right; 7, Left; 14			
The side of MAS		44.5 ± 7.2	41.3 ± 7.5	N.S.
RMT in more affected cortex (%) *		47.0 ± 7.2	46.3 ± 7.6	N.S.
RMT in less affected cortex (%) *		38.0 ± 5.9	36.9 ± 6.2	N.S.
AMT in more affected cortex (%) *		39.1 ± 6.2	39.0 ± 7.5	N.S.
AMT in less affected cortex (%) *				

* Data are shown as mean ± SD (range). LEED, levodopa equivalent daily dose; MDS-UPDRS, Movement Disorder Society Unified Parkinson’s Disease Rating Scale. RMT, resting motor threshold. AMT, active motor threshold. MAS, more affected side. N.S, not significant. H-Y, Hoehn and Yahr scale.

**Table 2 brainsci-12-00842-t002:** Correlation between the resting motor thresholds and the motor subscore in the ON and OFF periods.

	ON Period		OFF Period	
	More Affected Cortex	Less Affected Cortex	More Affected Cortex	Less Affected Cortex
MDS-UPDRS axial score	r = −0.352 *p* = 0.118	r = −0.360 *p* = 0.109	r = −0.425 *p* = 0.055	r = −0.302 *p* = 0.183
MDS-UPDRS MAS score	r = −0.181 *p* = 0.433	r = 0.014 *p* = 0.951	r = −0.439 *p* = 0.047	r = −0.075 *p* = 0.747
MDS-UPDRS LAS score	r = 0.007 *p* = 0.976	r = 0.161 *p* = 0.487	r = −0.119 *p* = 0.608	r = −0.012 *p* = 0.957

MDS-UPDRS, Movement Disorder Society Unified Parkinson’s Disease Rating Scale. RMT, resting motor thereshould. MAS, more affected side. LAS, less affected side.

## Data Availability

Data available on request due to restrictions eg privacy or ethical. The data presented in this study are available on request from the corresponding author.

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
