# Peer review of "The Lateralization of Resting Motor Threshold to Predict Medication-Mediated Improvement in Parkinson’s Disease"

_brainsci, 2022, doi:10.3390/brainsci12070842_

Round 1
Reviewer 1 Report
This brief report investigated whether TMS biomarkers, which are associated with cortical inhibitory and facilitatory processes, reflect disease severity in patients with PD. Although the results are interesting, some aspects of the manuscript should be better addressed. Please see my comments below.
Introduction:
Lines 37 to 42: some technical information appears here and again in the methods section (lines 84 to 88). As an alternative, the authors could improve this introduction section explaining how cortical function is translated into MEPs.
Please define what a TMS biomarker is and refer to the appropriate literature
The hypothesis of the study should be improved
Methods:
Line 85: please check “RMT (RMT)”
Please explain the rational for using Part 3 of MDS-UPDRS throughout the results
Results:
Table 1 is confusing and a better explanation on what each measure means would be helpful for the reader. Also, the p-values refer to which statistical test?
Figure 1: A boxplot would be more informative. Note that the authors state that outliers are shown as small circles, but it is not clear whether they were removed or not. Also, the authors indicated that “All data are presented as means ± SD”. Please, keep the charts consistent to what was indicated in the methodology.
Lines 131 and 132: please, expand the description on what is described in plates C and D.
Lines 136 and 137: the authors state that “The RMT of the affected side was correlated with the MDS-UPDRS part 3 in the OFF period.”. Please make it clear to the reader that this correlation was not significant according to the selected alpha level of 5%.
Figure 2 can appear after item 3.2
Section 3.3 (line 139): this section is confusing and needs improvement
Discussion
Lines 164 and 165: The sentence “Dopamine depletion in PD causes both increased concentrations of gamma-amino-butyric acid (GABA) in the thalamus and decreased facilitation of the motor cortex.” Is extremely vague and can be improved.
Please expand the limitations of this study, including the sample size.
Author Response
This brief report investigated whether TMS biomarkers, which are associated with cortical inhibitory and facilitatory processes, reflect disease severity in patients with PD. Although the results are interesting, some aspects of the manuscript should be better addressed. Please see my comments below.
Response: We wish to express our appreciation to the Reviewer for their insightful comments, which have helped us significantly improve the paper. Thank you for taking the time and energy to help us strengthen our paper.
Introduction:
Lines 37 to 42: some technical information appears here and again in the methods section (lines 84 to 88). As an alternative, the authors could improve this introduction section explaining how cortical function is translated into MEPs. Please define what a TMS biomarker is and refer to the appropriate literature
Response:Thank you for the insightful comment. We added the following content to Introduction: “Transcranial magnetic stimulation (TMS) is a valuable method to investigate the temporal dynamics of the motor system during perception of emotional faces, via stim-ulation of the primary motor cortex (M1) and the consequent induction of motor-evoked potentials (MEPs) in target muscles. The amplitudes of TMS-induced MEPs provide an instantaneous readout of the excitability of the corticospinal system, allowing us to probe distinct motor representations with high temporal resolution, and, importantly, to dis-tinguish between excitatory (MEP increase) and inhibitory (MEP decrease) motor processes [21].”
The hypothesis of the study should be improved
Response: Thank you for the insightful comment. We amended as follows: “We hypothesized that cortical inhibitory and facilitatory pathway functions affect the disease severity, the laterality of motor symptoms, and scope of response to medications administered for PD treatment.”
Methods:
Line 85: please check “RMT (RMT)”
Response: We thank the reviewer for pointing out the mistake. We have corrected this accordingly.
Please explain the rational for using Part 3 of MDS-UPDRS throughout the results
Response: We thank the reviewer for their insightful comment. We added the following content to the Methods: “Patients with PD were observed for one week prior to examinations and were assessed in the ON and OFF states, which correspond to states when dopamine replacement therapy is effective and symptoms are controlled and when the treatment effect wears off and symptoms re-emerge. The motor symptom examinations were performed for each patient during both the ON and OFF states over two testing days by the same examiner.”
Results:
Table 1 is confusing and a better explanation on what each measure means would be helpful for the reader. Also, the p-values refer to which statistical test?
Response: We amended Table1 as suggested.
Figure 1: A boxplot would be more informative. Note that the authors state that outliers are shown as small circles, but it is not clear whether they were removed or not. Also, the authors indicated that “All data are presented as means ± SD”. Please, keep the charts consistent to what was indicated in the methodology.
Response: We thank the reviewer for pointing out this mistake. We have corrected Figure 1, which is the methodology figure. We removed the confusing sentence “All data are presented as means ± SD”.
Lines 131 and 132: please, expand the description on what is described in plates C and D.
Response: Thank you for the insightful comment. We add the recommended explanation.
Lines 136 and 137: the authors state that “The RMT of the affected side was correlated with the MDS-UPDRS part 3 in the OFF period.”. Please make it clear to the reader that this correlation was not significant according to the selected alpha level of 5%.
Response: Thank you for the insightful comment. We added the recommended content to the Data Analysis of Methods Section.
Figure 2 can appear after item 3.2
Response: We thank the reviewer for pointing out this mistake. We have corrected the location.
Section 3.3 (line 139): this section is confusing and needs improvement
Response: We thank the reviewer for pointing out this mistake. We transferred this content to Section3.1.
Discussion
Lines 164 and 165: The sentence “Dopamine depletion in PD causes both increased concentrations of gamma-amino-butyric acid (GABA) in the thalamus and decreased facilitation of the motor cortex.” Is extremely vague and can be improved.
Response: Thank you for the insightful comment. We corrected the sentence as follows: “Dopamine depletion in PD causes not only an increase in concentrations of gamma-aminobutyric acid (GABA) in the thalamus, but also a decrease in the promoting effect of the motor cortex.”
Please expand the limitations of this study, including the sample size.
Response: We thank the reviewer for pointing out this mistake. We added the limitations to the Discussion section.
Reviewer 2 Report
This is an interesting study that evaluated transcranial magnetic stimulation parameters and correlated with severity and motor status in 21 patients with PD. The introduction is well written, as are the objectives of the study are well defined. The methods are well described, but:
1) the authors define the onset of the disease "upon the onset of muscle weakness". In my opinion this is not an adequate definition for the initial symptoms of PD.
2) the authors do not clearly describe how the procedure was to examine patients in the ON and OFF states.
The results are well presented, but:
3) the authors classify the body sides of PD patients as "affected" or "unaffected", when most have bilateral symptoms as described for disease stage by the Hoehn and Yahr scale. So in my opinion another description should be used, for example: "more affected side", and "less affected side".
4) The authors neither describe nor consider the dominant hand as a factor that could interfere with the observed results
5) The authors correlated the TMS parameters with the overall MDS-UPDRS motor scale scores, but I think they should also calculate the correlation of these parameters with the motor scores on each side of the body and with the axial symptom score.
I think the article is very good, but I think some corrections and a re-evaluation of the data are needed as pointed out above.
Author Response
This is an interesting study that evaluated transcranial magnetic stimulation parameters and correlated with severity and motor status in 21 patients with PD. The introduction is well written, as are the objectives of the study are well defined.
Response: We wish to express our appreciation to the Reviewer for their insightful comments, which have helped us significantly improve the paper. Thank you for taking the time and energy to help us strengthen our paper.
The methods are well described, but:
1) the authors define the onset of the disease "upon the onset of muscle weakness". In my opinion this is not an adequate definition for the initial symptoms of PD.
Response: Thank you for the constructive suggestion, with which we agree. We amended the definition of motor symptom onset. The time of appearance of non-motor symptoms is ambiguous. Therefore, we defined the time of appearance of motor symptoms.
2) the authors do not clearly describe how the procedure was to examine patients in the ON and OFF states.
Response: Thank you for pointing out insufficient descriptions. We added the sentence in Methods “Patients with PD were observed for one week prior to examinations and were assessed in the ON and OFF states, which correspond to states when dopamine replacement therapy is effective and symptoms are controlled and when the treatment effect wears off and symptoms re-emerge. The motor symptom examinations were performed for each patient during both the ON and OFF states over two testing days by the same examiner.”
The results are well presented, but:
3) the authors classify the body sides of PD patients as "affected" or "unaffected", when most have bilateral symptoms as described for disease stage by the Hoehn and Yahr scale. So in my opinion another description should be used, for example: "more affected side", and "less affected side".
Response: We have revised the terms to "more affected side" and "less affected side".
4) The authors neither describe nor consider the dominant hand as a factor that could interfere with the observed results
Response: All participants in this study were right-hand dominant, therefore we have added this limitation to the Discussion.
5) The authors correlated the TMS parameters with the overall MDS-UPDRS motor scale scores, but I think they should also calculate the correlation of these parameters with the motor scores on each side of the body and with the axial symptom score.
I think the article is very good, but I think some corrections and a re-evaluation of the data are needed as pointed out above.
Response: Thank you for pointing out the insufficient descriptions. We added additional analyses including each side of the body and with the axial symptom score. We added the section 3.3. “Correlation between RMT and motor subscores” and Table 2.
|
ON period |
OFF period |
|||
|
More affected cortex |
Less affected cortex |
More affected cortex |
Less affected cortex |
|
|
MDS-UPDRS axial score |
r=-0.352 p=0.118 |
r=-0.360 p=0.109 |
r=-0.425 p=0.055 |
r=-0.302 p=0.183 |
|
MDS-UPDRS MAS score |
r=-0.181 p=0.433 |
r=0.014 p=0.951 |
r=-0.439 p=0.047 |
r=-0.075 p=0.747 |
|
MDS-UPDRS LAS score |
r=0.007 p=0.976 |
r=0.161 p=0.487 |
r=-0.119 p=0.608 |
r=-0.012 p=0.957 |
Again, thank you for giving us the opportunity to strengthen our manuscript with your valuable comments and queries. We have made every effort to incorporate your feedback and hope that these revisions persuade you to accept our submission.
Reviewer 3 Report
In this manuscript, the authors present TMS stimulation on patients with Parkinson’s disease in order to investigate the corticospinal and intracortical excitability. The work is well written although some information need to be improved
Specific Comments: Authors should provide more details on the results about the active motor threshold intensity (AMT). Authors discuss about “change of RMT”: is it calculated as differences between right and left side? Please report the reference of literature.
Author Response
In this manuscript, the authors present TMS stimulation on patients with Parkinson’s disease in order to investigate the corticospinal and intracortical excitability. The work is well written although some information need to be improved
Specific Comments: Authors should provide more details on the results about the active motor threshold intensity (AMT). Authors discuss about “change of RMT”: is it calculated as differences between right and left side? Please report the reference of literature.
Response: We wish to express our appreciation to the Reviewer for their insightful comments, which have helped us significantly improve the paper. Thank you for taking the time and energy to help us strengthen our paper.
We added the analysis of AMT in the Results section. We remade Figure 2. In this study, the change in RMT indicated the change between ON and OFF states. We have added this explanation and some supporting references.
Again, thank you for giving us the opportunity to strengthen our manuscript with your valuable comments and queries. We have made every effort to incorporate your feedback and hope that these revisions persuade you to accept our submission.